# 'In fact, that's when I stopped using contraception': a qualitative study exploring women's experiences of using contraceptive methods in KwaZulu-Natal, South Africa

Mbuzeleni Hlongwa ![ORCID],[1,2] Chipo Mutambo,[1] Khumbulani Hlongwana[1,3]

¹School of Nursing and Public Health Medicine, University of KwaZulu-Natal, Durban, South Africa
²Public Health, Societies and Belonging programme, Human Sciences Research Council, Pretoria, South Africa
³Cancer & Infectious Diseases Epidemiology Research Unit, University of KwaZulu-Natal, Durban, South Africa

**Correspondence to**
Mbuzeleni Hlongwa;
hlongwa.mbu@gmail.com

## ABSTRACT

**Objectives** This study explored women's experiences of using contraceptive methods in KwaZulu-Natal, South Africa.

**Setting** In October 2021, we conducted a qualitative study at Umlazi Township in KwaZulu-Natal province, South Africa, through face-to-face in-depth interviews.

**Participants** Women from four primary healthcare facilities were recruited through a criterion-based sampling strategy. Using NVivo V.11, two skilled researchers independently conducted thematic data analysis, as a mechanism for quality assurance, before the results were collated and reconciled.

**Results** The study included 15 female participants, aged between 18 and 35 years, of whom two-thirds were aged 18–24 years. We found that women were concerned about unpleasant contraceptive methods side effects such as prolonged or irregular menstrual periods, bleeding, weight gain and/or severe pains, resulting in discontinuation of their use. In addition to contraceptive stockouts, women indicated that healthcare providers did not appropriately counsel or inform them about the available contraceptive methods, including how to use them. Key themes included the following: negative effects of contraceptive methods; stockouts of preferred contraceptive methods; inconsistent or incorrect use of contraceptive methods; lack of counselling regarding contracepive methods; and misconceptions about contraception.

**Conclusions** Interventions aimed at reducing contraceptive stockouts are required to ensure that women are empowered to choose contraception based on their own preference, convenience and/or experience. It is imperative that counselling on contraceptive methods' side effects be improved, to ensure that women have freedom to make informed decisions about their preferred method, proper management of side effects and to assist them with method switching as needed, instead of discontinuation.

## INTRODUCTION

Contraception is one of the most important public health interventions which respond to sexual and reproductive health needs of women, thereby enabling them to plan their pregnancies and decide on the number of

## STRENGTHS AND LIMITATIONS OF THIS STUDY

⇒ Use of in-depth qualitative interviews elicited rich data that offer important insights into sexually active women's discontinuation of contraceptive use.
⇒ Given that this study focused on women attending public healthcare facilities, women who do not frequently use public healthcare clinics may have been excluded, hence their insights would be missing.
⇒ The findings of this study were sought from participants' self-reported sexual and reproductive health information, rendering them prone to social desirability bias.
⇒ Given the sensitivity of the topic, information prone to moral judgements, may have been withheld by the participants for image preservation.

children they desire.[1] The consistent and correct use of contraception has far-reaching benefits to both individuals and societies, including the reduction of pregnancy-related morbidity and mortality, termination of pregnancies (ToPs), improving educational opportunities and empowerment of women.[2 3] Furthermore, contraceptive use positively affects the overall health of women of reproductive age, as it empowers them to autonomously make decisions regarding their own sexual and reproductive health.[4 5] Studies have shown that empowered and educated women are more likely to make informed decisions about the use of contraception.[6 7]

Despite the notable improvements in the uptake of contraceptive methods by women of reproductive age globally, the sub-Saharan African (SSA) region continues to record a high proportion of women who experience unplanned pregnancies every year.[8] In SSA, unmet contraception needs for women of reproductive age continue to be a public health concern, with a substantial number of these women being unable to access and

use their preferred methods of contraception, which jeopardises their desire to circumvent unplanned pregnancies.[9 10] We defined an unmet need for contraception as the proportion of women who are either (A) not pregnant, (B) not postpartum amenorrhoeic, are considered fecund, want to delay their next pregnancy or (C) are postpartum amenorrhoeic and their most recent birth within the previous 2 years was unwanted or mistimed, but they are not using a contraceptive method.[11 12] The rate of unmet needs for contraceptive use is more than 20% among sexually active women in SSA.[13 14]

The 2016 Demographic and Health Survey report showed that 55% and 60% of married women and sexually active women, respectively, used contraceptives in South Africa.[11] At least 15% of women who are in-union, have an unmet need for contraception to prevent unplanned pregnancy, and the figure is slightly higher for sexually active women (19%). However, the KwaZulu-Natal province accounts for 18% and 21% of the national figures of unmet need for contraception, respectively, among in-union and sexually active women.[11] In South Africa, more than 100 000 terminations of pregnancy were reported in designated facilities in the 2016/2017 financial year, with the KwaZulu-Natal province accounting for 15% of these.[15] Despite ToPs being legal in South Africa through the Choice of Termination of Pregnancy Act, which was made public in 1996, and amended in 2008,[16] many ToPs are conducted illegally, contributing to high maternal morbidity and mortality rates.[3 17 18] The aim of the act is to reduce maternal mortality from illegal ToPs, and permits pregnant women to terminate their pregnancy legally.[16]

High rates of unplanned pregnancy have also been reported among women diagnosed with HIV in South Africa.[19 20] Risky sexual behaviours are also common among young women in this country, as well as in KwaZulu-Natal, including early sexual debut, sex under the influence of alcohol or drugs, multiple sexual partners, and inconsistent condom use, all of which contribute to high levels of unplanned pregnancy and sexually transmitted diseases, including HIV.[21–26]

Factors contributing to high rates of unplanned pregnancy in resource-limited settings, including South Africa, are well documented. Insufficient knowledge of contraception, gender inequality, intimate partner violence, poverty, and inconsistent and incorrect use of contraception, are some of the factors.[8 10 27 28] Low levels of contraceptive use have been associated with age, low education level and low socioeconomic status, limited knowledge and inaccessibility of contraceptive methods, and resource-limited rural residential settings.[10 23 29–33] Among adolescent girls, some of the barriers to contraceptive use include the lack of desire to avoid, delay, space or limit childbearing; the lack of confidence and ability to seek and/or negotiate contraceptive use; the lack of desire to use contraception; and poor access to contraceptive services and to quality.[34] Health system challenges, including contraception stockouts, long waiting times and negative attitudes displayed by some healthcare providers, have also been reported to deter women, especially young women, from accessing contraceptive services from healthcare facilities, thereby contributing to low or inconsistent uptake of contraceptives.[32 35–37] High rates (29%) of contraceptive users discontinue using contraception within a year of starting to use contraception, due to side effects (28%), the desire to become pregnant (19%) or the desire for a more effective method (11%).[11] Contraceptive stockouts remain high in South Africa, deterring women from accessing preferred contraceptive methods in local public healthcare facilities, thereby contributing to low or inconsistent uptake of contraceptives.[11 38 39]

Despite these challenges, the South African government has demonstrated commitment to ensuring universal access to contraception by women of reproductive age. In South Africa, only 46% of births occurring within a 5-year period preceding the South African Demographic and Health Survey 2016 were intended at the conception time, while 34% and 20%, respectively, were mistimed and unwanted.[11] This study explored women's experiences of using contraceptive methods in KwaZulu-Natal province, South Africa. The use of qualitative research methods deepened our understanding of this phenomenon.

## METHODS
### Study setting
Umlazi, a township populated with more than half a million people, is located in the province of KwaZulu-Natal[40] and is part of the eThekwini Metropolitan Municipality, which has the largest number of people on lifelong antiretroviral therapy (N=450 238) in the province, and accounts for 16.7% of HIV prevalence.[41] Umlazi township has 10 public healthcare clinics and one public health hospital. Four public healthcare clinics, spread across various parts of Umlazi, participated in the study.

### Study design
We explored women's experiences of using contraceptive methods, using an exploratory descriptive qualitative study design.[42] We generated data in October 2021, through face-to-face in-depth interviews with women of reproductive age accessing healthcare services, from four facilities in Umlazi.

### Sampling
We employed criterion-based sampling strategies, to identify potentially eligible women from four primary healthcare facilities. In order to be eligible for selection, women had to meet the following criteria: (A) be of reproductive age (18–49 years); (B) reside in Umlazi Township; (C) be sexually active (women who had sexual intercourse within 3 months preceding data collection) and (D) use contraception or would have used contraception within 3 months preceding data collection. Women who were outside the age brackets (18–49 years), pregnant or sexually inactive were excluded from the study. Interviews

were conducted after services had been rendered by the facility, in a designated quite area within the facility.

## Data collection tool

We developed an in-depth interview guide in the English language and then translated it into IsiZulu, which is the dominant language used in Umlazi. A female research assistant with a track record of conducting interviews and qualitative research, was recruited and provided with refresher training prior to collecting the data. The research assistant is fluent in both English and IsiZulu languages. Interviews were audiorecorded (with participants' permission). To minimise any inconsistencies during the data collection period, the interview guide was pretested with five participants who did not form part of the study. The scope of inquiry of the interviews focused on four key components of contraception, namely: (1) awareness about different contraceptive methods, (2) access (availability/stockouts of preferred contraceptive methods, counselling), (3) uptake (key considerations in deciding whether or not to start using contraception) and (4) adherence/continuation (key considerations in deciding whether or not to continue or discontinue using contraception, including side effects, violence, stigma, discrimination, judgements).

## Data collection

Data were collected iteratively to ensure that researchers engaged with preliminary data analysis of the information collected, learn emerging themes, identify gaps in the data and adapt the data collection process for subsequent interviews. Each interview between 30–60 min. Data saturation was reached at 15 interviews. At least five potential participants refused to participate in the study when they were approached. The lead investigator held regular debriefing meetings with the research assistant to discuss field experiences, challenges, emergent issues, lessons learnt, and how she (Research Assistant) affected and/or was affected by the interviews. Data collection only took place after the services had been rendered by the facility. These participants were recruited regardless of the service(s) they presented themselves for at the facility. We did not collect information regarding the care service they visited the facility for.

## Data analysis

Using NVivo V.11, two researchers independently conducted the analysis from data coding to the development of themes, and this was done iteratively, guided by Richie and Spencer's framework.[43] The framework outlines the following stages for conducting qualitative data analysis: (A) familiarisation with the data through reading all the transcripts and listening to the audio recordings; (B) generating initial codes using an open coding method, where each segment of data relevant to this study's research objective was coded; (C) development of a thematic framework extracting key themes from the coded data; (D) application of the thematic framework

| Table 1 Demographic characteristics of participants, Umlazi township, KwaZulu-Natal, South Africa, 2021 ||
|---|---|
| **Characteristic** | **Participants n=15** |
| Age (years) | 18–49 |
| Median (IQR) | 23 (19–26) |
| Age categories, n (%) | |
| 18–24 years | 10 (67) |
| 25–49 years | 5 (32) |
| Population group, n (%) | |
| Black African | 15 (100) |
| Level of education, n (%) | |
| Secondary | 12 (80) |
| Tertiary | 3 (20) |
| Employment status, n (%) | |
| Unemployed | 7 (46) |
| Employed | 4 (27) |
| Studying | 4 (27) |
| Marital status, n (%) | |
| Not married | 14 (93) |
| Married | 1 (7) |

to all the data; (E) charting of the data, enabling systematic comparisons between data sets and (F) analysis of the charts for patterns and associations between and within each unit of analysis.

## Data quality

To ensure rigour and accuracy, comparative data analysis was conducted by two skilled researchers, who independently read all the transcripts to gain an understanding of the content and scope of the data collected, prior to data coding. The outcome of coding was verified, cross-checked and thoroughly discussed between the two members, to ensure that the research question was answered. We used the COnsolidated criteria for REporting Qualitative research (COREQ) checklist to ensure that the study adhered to quality standards for reporting qualitative study findings.[44]

### Patient and public involvement

Patients or the public were not involved in the design, or conduct, or reporting, or dissemination plans of our research.

## RESULTS
## Background characteristics of study participants

The study included 15 women, aged between 18 and 35 years, with a median age of 23 years (table 1). All participants were black African women, with more than two-thirds (n=10) aged 18–24 years. The majority (n=12) had attained a secondary level of education and nearly half (n=7) were unemployed. Almost all participants (n=14) were not married.

## Key themes

The following key themes were developed during analysis: negative effects of contraceptive methods; stockouts of preferred contraceptive methods; inconsistent or incorrect use of contraceptive methods; lack of counselling regarding contraceptive methods and misconceptions about contraception.

## Negative effects of contraceptive methods

Some participants reported unpleasant side effects resulting from using some contraceptive methods. The most common side effect reported by the participants, was prolonged menstrual periods resulting from using injectable contraceptives.

*The injection made me have prolonged periods than usual. I usually go on periods for three days, but after taking the injection, my periods would last for seven days… I was tired of being on periods all the time. My periods lasted longer than usual. I'd keep having periods. Then it would stop eventually, but then again the following months when it's time for my periods, they would last longer… The next month, same thing.* (19-year-old)

*I went on a prolonged menstrual cycle, my periods started from the 2nd to the 12th of August… I don't know what is going on, I don't know…But the injection didn't treat me well. I had started with the two months injection. But I stopped using the two months injection because it would make me go on periods at any time. It made me bleed at any time.* (19-year-old)

Some women indicated that using the injectable contraceptive method made them bleed heavily, or that the injectable method interfered with their menstrual cycle and resulted in inconsistent menstrual periods.

*She [nurse] had given me the 3 months injection. The three months injection was making me bleed a lot. My bleeding couldn't stop. Then I did some research and asked from others and they said that it happens that the injection would not respond well on your body".* (20-year-old)

*It was not treating me well. It would make me go on periods at least twice in one month. I would go on periods for three or four days but twice a month. In the beginning it treated me well. Then it changed later on.* (25-year-old)

Women reported that using contraception contributed to changes in their appetite for food, thereby affecting their weight. Some experienced weight gain while others lost weight.

*…when I started with the contraceptives, I gained a lot of weight and I was oily, like my face was messed up so I stopped and then after a year I went back [re-started contraception]….* (22-year-old)

*I started by using the three months injection, but it made me lose weight so bad, then I switched from it to the two months injection.* (25-year-old)

Some participants reacted to side effects by simply discontinuing contraception, while others were forced to switch to a different contraceptive method.

*The injection made me have extended periods than usual… So I didn't even ask, I just decided to stop using contraception. In fact, that's when I stopped using contraception. I stopped using contraception last year or last-of-last year.* (19-year-old)

*Then I came back and told the nurse about the side effects I was experiencing while using that injection. I then asked her to switch me to a different method. And then she switched me to a two-months injection.* (20-year-old)

## Stockouts of preferred contraceptive methods

Participants reported stockouts of their preferred contraceptive methods as a pervasive challenge in the public healthcare facilities.

*Yes, even last week they said they were out of stock, especially the three months injection, they only had the two months one.* (18-year-old)

*…sometimes you come here and find that the injection is not available… I don't know what they can do to improve the situation. I sometimes ask the nurses why they don't have the injections and they say that it is the department that is not delivering them.* (20-year-old)

Changing to an alternative contraceptive method was one of the options pursued by some participants whenever they were unable to receive their preferred contraceptive method, while others opted to wait for their preferred contraceptive method. Those who opted to wait for their contraceptive method of choice, had to modify their lifestyle during the period of being not protected by contraception.

*I continued using the three months injection. But it would happen that sometimes they would not have it at the clinic. Then I would be switched to the two months injection.* (20-year-old)

*I decided to stop. I didn't take them. I stopped using contraceptives. My date was on the 12th September. I came here and I was told that the injection is not available. Then, again I came back on the 18th September. And they told me again that the injection is not available. Then I waited. Then I received a message from my friend who told me that she was at the clinic and that the injection is now available. I think it was last week. Then I came back and took the injection.* (20-year-old)

*For me I don't want to use something else when I'm still going to use the implant. I don't want too many in my body. At least if I use one. I should not mix. I should just be patient for the implant. I should be able to control myself. For now, I'm trying to avoid having sex, since August.* (24-year-old)

The contraceptive methods stockouts was reported to have far-reaching implications on women, as it does not

only make women vulnerable to unplanned pregnancies, but it also cause emotional discomfort.

*During that time the injection is not available, you are not comfortable. You are scared. It is hard when the injection is not available.* (20-year-old)

*You'll end up getting discouraged if you don't get the contraceptive methods you need. All they need to do is to ensure that the preferred contraceptive methods are always available.* (35-year-old)

### Inconsistent or incorrect use of contraceptive methods

Participants indicated that they sometimes could not consistently use contraceptives, as some of them missed their reinjection appointments. This resulted in some of them experiencing an unplanned pregnancy.

*…When my re-injection date had passed, as I was taking a break, so that my blood will flow and the side effects would stop, I then got pregnant. I had planned to wait for a month. Then I would start the following month. It was stupid I don't know how [laughs]. And then during that period I became pregnant. It just happened so quickly. I couldn't believe it. I couldn't believe that I was really pregnant.* (23-year-old)

*I do have a child. I didn't stop using contraception, but I got the child after I skipped my date for re-injection. It just happened that I got pregnant.* (25-year-old)

Most participants showed less preference for the contraceptive pills, fearing that they would not be able to maintain or use them consistently or correctly.

*…what made me stop using them [contraceptive pills] is because I used to miss them because you have to take them at the same time daily, I stopped taking them… I stopped using them because I kept missing the time whereas you are supposed to take them at the same time.* (19-year-old)

*I've never used something else. I've only used the injection. I cannot use the pills because I would forget them. I forget the pills I cannot use them.* (23-year-old)

Some participants reported instances where healthcare providers either inserted the implant incorrectly, or inserted it into a pregnant woman, as some healthcare providers hardly ever checked a woman's pregnancy status before inserting the implant.

*…and I also got pregnant while the implant was inserted in me… Maybe they inserted it while I was already pregnant. They didn't check my pregnancy status. They didn't run pregnancy tests. They just inserted it.* (19-year-old)

*I had pregnancy symptoms. I would vomit. I would experience things that I didn't understand. Then I went back to the clinic and I was told that the implant was not inserted correctly. Then they started telling me about the possible risks since I am already pregnant. That I might get a miscarriage. And I did get a miscarriage. I don't know if that's what caused the miscarriage. After they removed the implant, I had already lost hope because I was bleeding.* (19-year-old)

### Lack of counselling regarding contraceptive methods

Participants indicated that they were not counselled or given adequate information regarding the use, and the possible side effects, of various contraceptive methods offered at healthcare facilities.

*…the nurse did not explain anything to me. She did not tell me how it works. I do have some questions about using contraception. I want to ask that since they are different, there's one for three months and another one for two months – what's the difference between them? That's all I want to ask.* (18-year-old)

*She didn't even ask me which method I wanted to use or what. She didn't and I couldn't ask questions. I didn't even know which injection she was giving to me [after giving birth at the hospital]. She didn't say anything. She just calls your name and then gives you the injection… It got me confused. Because I wanted to use a different method. But I only realized what they had given me when I was reading the card because it was written there. And there was nothing I could still do.* (20-year-old)

*…there was no opportunity because in the room we were getting injections from, we were many so there was no space for discussing or asking questions. So you'd get the injection and go home. You didn't have time to discuss things or ask questions.* (24-year-old)

### Misconceptions about contraception

Concerns over the effectiveness of using contraceptive methods were raised, with some women reporting that they do not trust that the contraceptives are effective enough to prevent someone from getting an unplanned pregnancy.

I think that because there's many different contraceptive methods, what I know about the injection is that it reduces chances of getting pregnant. But it is not 100% accurate…. I will continue using a condom because I know that the injection is not 100% sure. (18-year-old)

Some participants raised concerns about the possibility of contraceptive method impairing future fertility, with others preferring to wait until they got their first child before using contraceptive method.

*…but the very first time I heard about contraception, I was told by my mother who asked me to use contraception, so that I won't get pregnant. I hadn't had a child at that time. But I refused because I heard people saying that if you use contraception when you haven't had a child, it happens that you might not be able to conceive. So I didn't use contraception. I only started using contraception after getting a baby. I have one child.* (20-year-old)

*I only used it for a short period because I was told that it's dangerous to use the injection when you don't have a child because it completely prevents you from getting pregnant in future…..* (25-year-old)

## Discussion

In this study, we adopted a qualitative approach to exploring sexually active women's experiences of using contraceptive methods in KwaZulu-Natal, South Africa. Our study participants shared crucial insights for understanding the challenges women face, when accessing and using contraceptive methods in Umlazi township, KwaZulu-Natal. Our results show that negative effects of contraceptive methods; stockouts of preferred contraceptive methods, inconsistent or incorrect use of contraceptive methods; lack of counselling regarding contraceptive methods; and misconceptions about contraception, affect the use of contraceptives in the study setting, and this may have some relevance for other comparable settings.

Side effects play an important role in a woman's decision as to whether to continue or discontinue using contraception, as has been reported in other studies.[45 46] Similar to our study, prolonged or irregular menstrual cycle patterns have also been reported as one of the most common side effects that influence women to discontinue contraceptive use.[46–48] Both the injectable users and implant users have concerns about bleeding side effects, which are known to have contributed to discontinuation of contraception in similar settings.[46] Contraceptive discontinuation has dire implications, including unplanned pregnancy. Therefore, women should be empowered to avoid unplanned pregnancies, by identifying alternative contraceptive methods that meet their individual needs.

There have been concerns about the growing number of women who return to clinics for implant contraceptive removal, only after a few months of insertion, due to changes in bleeding patterns or excessive bleeding.[47 49 50] Side effects, notably prolonged or irregular bleeding, have also been reported by healthcare providers as the most common reason for early contraceptive implant discontinuation.[47] In addition to this, other factors have been reported to influence women to remove the implant, including incorrect positioning and low quality of care.[51–53] Incorrect positioning of the implant was also reported in this study as an important factor for its discontinuation. The contraceptive implant protects against pregnancy for up to three years before it needs to be replaced, and this reduces frequent clinic visits among users.[54] However, the growing number of women removing the implant given the side effects, suggests the need for preinsertion counselling, proper management and empowerment of women through information regarding alternative contraceptive options for better decision-making.[50]

Contraceptive stockouts occur when one or more contraceptive methods are not available at a healthcare facility that routinely provides that method. Contraceptive methods stockouts are one of the main factors contributing to contraceptive switching and discontinuation among women.[55 56] Contraceptive methods should be available at all levels of healthcare, but not all healthcare facilities offer the full mix of modern contraceptive methods. Women have different contraceptive method preferences and experiences; therefore, a wide variety of contraceptive methods should always be available; to ensure that women are empowered to choose contraception based on their own preference, convenience and/or experience. It has been reported that the use of modern contraception increases when more methods become available.[57] Given the high rates of contraceptive method stockouts in South Africa, unemployed women are likely to suffer the most, due to financial constraints.

Healthcare providers play an important role in ensuring access to contraceptive methods among women. As such, healthcare providers have an important responsibility in educating women and providing complete information about possible side effects, and the effectiveness of their preferred contraceptive method. It is imperative that counselling on side effects be improved, to ensure that women have freedom to make informed decisions about their preferred method, and to assist them with method switching as needed. However, counselling when receiving family planning services is limited in South Africa.[39] Moreover, a single face-to-face counselling appointment in a busy facility may not be enough to convey all of the information a woman requires about: the reproductive cycle, returning to fertility after discontinuing a method, potential drug–drug interactions, the need for dual protection and so on.[39] In addition to traditional face-to-face interactions between healthcare providers and contraceptive users, innovative approaches, such as internet-based sources of information, text message reminders and brochures are needed to improve women's comprehension of how contraceptive methods work.[39 58 59]

Our study has important limitations to note. The findings of this study were sought from participants' self-reported sexual and reproductive health information, rendering them prone to social desirability bias. Given the sensitivity of the topic, information deemed to have the potential for leading to judgements, may have been withheld by the participants for image preservation. Given that this study focused on women attending public healthcare facilities, women who do not frequently use public healthcare clinics may have been excluded, hence their insights would be missing. Furthermore, the study planned to include participants aged 18–49 years. However, we strangely did not get women who were aged 36–49 years. In retrospect, this is something that we could have interrogated further. Lastly, due to the qualitative nature of the study, the fact that this study was confined to limited healthcare facilities and participants, meant that our study findings cannot be generalised to other settings. However, the study provides important insights regarding the experiences of contraceptive use among sexually active women in Umlazi township, KwaZulu-Natal, South Africa.

## Conclusion

Our findings illustrate that concerted efforts are urgently required to address women's concerns regarding the side effects arising from using contraceptive methods,

as well as contraceptive stockouts, given the dire implications these may have on contraceptive discontinuation and subsequent unplanned pregnancy. The provision of comprehensive counselling services to support women who are having short-term side effects is paramount, to ensure that they can deal with side effects, or switch to a different method instead of completely discontinuing contraceptive use, to avoid unplanned pregnancy.

**Acknowledgements** The authors would like to thank the School of Nursing and Public Health, University of KwaZulu-Natal, Durban, South Africa and the Division of Research Capacity Development at SA MRC.

**Contributors** MH conceptualised and designed the study, as well as prepared the initial draft. KH and CM reviewed the manuscript. All the authors approved the final version of the manuscript. The author responsible for the overall content as the guarantor in this article is MH. The author accepts full responsibility for the work and/or the conduct of the study, had access to the data, and controlled the decision to publish.

**Funding** The work reported here was made possible through funding by the South African Medical Research Council through its Division of Research Capacity Development under the Bongani Mayosi National Health Scholars Programme from funding received from the South African National Treasury.

**Disclaimer** The content hereof is the sole responsibility of the authors and does not necessarily represent the official views of the SAMRC or the funders.

**Competing interests** None declared.

**Patient and public involvement** Patients and/or the public were not involved in the design, or conduct, or reporting, or dissemination plans of this research.

**Patient consent for publication** Consent obtained directly from patient(s)

**Ethics approval** Ethics approval and gatekeeper permission were obtained from the University of KwaZulu-Natal Biomedical Research Ethical Committee (BREC) (Ref No: BE424/18) and the Department of Health's National Health Research Database (NHRD) (Ref No: KZ_2018009_013), respectively. The eThekwini District's Ethical Review Committee also approved the study. The study was also supported by the participating health facilities. All participants who volunteered to participate in the study signed an informed consent form prior to their participation. The privacy and confidentiality of participants were protected.

**Provenance and peer review** Not commissioned; externally peer reviewed.

**Data availability statement** Data are available on reasonable request.

**ORCID iD**
Mbuzeleni Hlongwa http://orcid.org/0000-0002-5352-5658

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
