## [Reviewer comments · BMJ Open]

ARTICLE DETAILS

TITLE (PROVISIONAL)	"In fact, that's when I stopped using contraception": A qualitative study exploring women's experiences of using contraceptive methods in KwaZulu-Natal, South Africa
AUTHORS	Hlongwa, Mbuzeleni; Mutambo, Chipso; Hlongwana, Khumbulani

VERSION 1 – REVIEW

REVIEWER	Khan, HR University of Dhaka
REVIEW RETURNED	23-Jun-2022

GENERAL COMMENTS	This is a qualitative study but the alternative quantitative study with sufficient data could address the objectives much more precisely. The objective is nice but the qualitative study with 15 interviews does not support enough evidence for the objectives and hence suffers from the lack of merits. The references are written inconsistently. It is unclear what "combination of convenience and criterion-based sampling techniques" means. No participants of >35 age were taken, why?
---

REVIEWER	Pleaner , Melanie University of the Witwatersrand Johannesburg Faculty of Health Sciences
REVIEW RETURNED	17-Aug-2022

GENERAL COMMENTS	Line 32: abstract – Not sure what is meant by this, already referred to side effects, but what potential negative effects are being referred to???? – re-word- "potential negative effects of various contraceptive methods" Page 5 lines 34-39 – sentence needs editing and unpacking, ideas and references are conflated, referencing needs to be aligned by the point: "Studies have shown that high rates of contraceptive users discontinue using contraception within a year of starting to use one of the methods, while others switch to a different method due to side effects, as well as stock-outs of preferred methods in local public health care facilities (10, 31, 32). Discontinuation is an important them and should be explained. Method switching perse is not a problem, especially if a woman experiences side effects, so should be dealt with as a separate issues. Stock outs is a challenge in local health facilities and should be referenced, and explained why it's a problem (in terms of SRH rights and choice). Reference 31 is out of date and more recent articles should be cited. Page 5 line 45: consider using the word were planned or intended – "...wanted at the conception time,"
---

Page 5 line 54 – edit sentence -clumsy “...this will in turn guide the development and/or the strengthening of interventions ...

Lit review, findings and discussion

Insights into HIV and barriers to contraceptive use in more detail, given HIV in KZN (see for example,

High incidence of unplanned pregnancy after antiretroviral therapy initiation: findings from a prospective cohort study in South Africa

SR Schwartz, H Rees, S Mehta, WDF Venter, TE Taha, V Black

PloS one 7 (4), e36039; 158

2012

An urgent need for integration of family planning services into HIV care: the high burden of unplanned pregnancy, termination of pregnancy, and limited contraception use among ... S Schwartz, E

Papworth, M Thiam-Niangoin, K Abo, F Drame, D Diouf, ...JAIDS Journal of Acquired Immune Deficiency Syndromes 68, S91-S98

Lit review: recommend use of some additional updated and relevant key sources:

A very good source on patterns of use in SA is: Chersich MF, Wabiri N, Risher K, Shisana O, Celentano D, Rehle T, Evans M, Rees H. Contraception coverage and methods used among women in South Africa: A national household survey. S Afr Med J. 2017 Mar 29;107(4):307-314. doi:

10.7196/SAMJ.2017.v107i4.12141. PMID: 28395681.

Jonas K, Duby Z, Maruping K, Dietrich J, et al.: Perceptions of

contraception services among recipients of a combination HIV-prevention interventions for adolescent girls and young women in South Africa: a qualitative study. Reprod Health. 2020; 17 (1): 122

PubMed Abstract | Publisher Full Text

Lit review, findings and discussion – any insights into barriers to

contraception, other than stock outs, especially for young people? E.g. Chandra-Mouli V, Improving access to and use of

contraception by adolescents: What progress has been made, what lessons have been learnt, and what are the implications for

action? Best Pract Res Clin Obstet Gynaecol. 2020 Jul;66:107-118. doi: 10.1016/j.bpobgyn.2020.04.003. Epub 2020 Apr 24.

PMID: 32527659; PMCID: PMC7438971.

Page 6 line 20: Study design – the study intended to explore why women discontinued on contraception, NOT as you state “We

explored women’s experiences of contraceptives in relation to their sexual behaviour” this needs clarification

Page 6 line 41 suggest delete “or not”

Page 9 line 37 risky sex – the theme is more about unplanned,

spontaneous sex rather than risky behaviour. Also deals with condomless sex rather than use of contraception. Can also raise

this in the discussion -concerning dual protection.

Page 14 line 42 edit sentence “The contraception stock out ...”

suggest Contraceptive stock outs

Page 17 line 24 – this deals with condomless sex and not

necessarily contraception

Some references you may wish to consider using in relation to missed appointments for injectable :

Baumgartner JN, Morroni C, Mlobeli RD, Otterness C, Myer L, Janowitz B, Stanback J, Buga G. Timeliness of contraceptive

reinjections in South Africa and its relation to unintentional discontinuation. Int Fam Plan Perspect. 2007 Jun;33(2):66-74. doi:

10.1363/3306607. PMID: 17588850.

Stockouts and importance of choice see Ross and Stover se of modern contraception increases when more methods become

available..

	Check and verify your definition of unmet need: “Unmet need -DHS Proportion of women who (1) are not pregnant and not postpartum amenorrhoeic and are considered fecund and want to postpone their next birth for 2 or more years or stop childbearing altogether but are not using a contraceptive method, or (2) have a mistimed or unwanted current pregnancy, or (3) are postpartum amenorrhoeic and their most recent birth in the last 2 years was mistimed or unwanted” -DHS 2016.
--	---

REVIEWER	Larsson, Elin C. Karolinska Inst
REVIEW RETURNED	29-Sep-2022

GENERAL COMMENTS	September 27th 2022 Review bmjopen-2022-063034 IN GENERAL  ○ Review the english throughout the manuscript References used are old, there has been a lot published on this topic recently ABSTRACT  ○ The English is not correct, pls rephrase the objectives that now reads: “Objectives: This study explored the experiences of women of reproductive age in the use of contraceptives, in relation to their sexual behaviour in KwaZulu-Natal, South Africa.” ○ Participants- what type of vcare had they sought at the health facilities from where they were recruited? INTRODUCTION  ○ Page 4, Line 51- move the (19%) after “sexually active” ○ Page 4, Starting line 47, make this into 2 meanings- difficult to follow: “At least 15% of women who are in-union, have an unmet need for contraception to prevent unplanned pregnancy in the country, and the figure is slightly higher (19%) for sexually active women; meanwhile, KwaZulu-Natal (KZN) accounts for 18% and 21% of the national figures (unmet need for contraception), respectively, among in-union and sexually active women (10).” ○ Page 4, Add information re legality of abortion I SA and KZN ○ Page 5, line 17- change “limited resource settings”, to resource limited settings ○ Page 5, Line 17-39- refers to SA or globally? ○ Page 5, line 51-, change qualitative methods to qualitative research methods ○ Page 5, a much clearer aim is needed. Now it reads “The use of qualitative methods to explore women’s experiences of contraceptives will deepen our understanding of this phenomenon; this will in turn guide the development and/or the strengthening of interventions aimed at improving contraceptive use among women of reproductive health in KwaZulu-Natal, and other comparable resource-limited
---

settings.”. Perhaps: “This study will use qualitative research methods to explore womens experiences of contraceptive use and reasons for non-use? And/or discontinuation? “

○

METHODS

- Page 6, explain what “EThekwini Metro” is.
- Page 6, you write about “eligible women” but ahvent specified the eligibility criteria? Why were the women at the health facility? Did you recruit them from a specific clinic or from the whole facility?
- Page 6, data collection tool- pls mention the topics covered in that.
- Add more information about where the data collection took place. At the facility? Directly upon recruitment or later etc.

RESULTS:

- **In general, there is an unbalance between text and quotes, which makes me think that the analysis is not thoroughly done. I highly recommend to re-write and collapse some themes. And select fewer quotes and instead add descriptive and summarizing text before ethe quotes.**
- Page 9, I would recommend to change the following sentence to “The following key themes emerged”, The following key themes were **developed** during analysis
- The themes should be rephrased so that they are related to the aim of the study (the aim I have suggested to revise to make it more precise in the intro)
- First section of “Early sexual debut and fear of unplanned pregnancy” doesn’t need 3 different quotes, I suggest to remove the last one.
- I would start the first theme with unplanned pregnancies, as these results are not about contraceptive use, and the contrast this to some women who started w contraception when they also started to have sex.
- Also I don’t see the “fear” coming out in the text under the same theme
- The theme “Participants’ reactions to side effects” is very short and almost only contains quotes. I would group these findings with the theme “Concerns over contraceptives’ efficacy and their side effects”
- Theme: “Lack of counselling on contraceptive methods” needs to be grouped with another theme, it is too short.
- I would group “Concerns over contraceptives’ efficacy” and “Misconceptions about contraception” into the same theme
- The theme “Spontaneity of sexual activities“, seems to be outside the scope of this paper

DISCUSSION

- **In general the discussion should be shortened and worked through, as it is going through the results each by each, could some be combined, the important aspects synthesized?**
- First sentence in dicsussion reads “Understanding the disparities in contraceptive use in South Africa, requires a thorough investigation of factors contributing to contraceptive uptake and discontinuation”, if this is the aim

	of this paper, pls review and adjust the aim in intro and the results accordingly.  ○ Page 19, line 50 and onwards, pls link and discuss the results to other studies. ○ The section on limitation could be expanded on. Nothing on trustworthiness etc. ○ Conclusion could be shortened, and the details of the study setting be removed
--	---

VERSION 1 – AUTHOR RESPONSE

(d) use contraception or would have used contraception within three months preceding data collection...'	Page 6, lines 3-8	
 • No participants of >35 age were taken, why? 	The study planned to include participants aged 18-49 years. However, we strangely did not get women of the said age group. In retrospect, this is something that we could have interrogated further.	
Reviewer 2	Authors responses	
 • Line 32: abstract – Not sure what is meant by this, already referred to side effects, but what potential negative effects are being referred to???? – re-word- “potential negative effects of various contraceptive methods” 	This sentence has been revised as follows: ‘Additionally, women indicated that health care providers did not appropriately counsel or inform them about the available contraceptive methods and how to use them’	Page 2, lines 32-33
 • Page 5 lines 34-39 – sentence needs editing and unpacking, ideas and references are conflated, referencing needs to be aligned by the point: “Studies have shown that high rates of contraceptive users discontinue using contraception within a year of starting to use one of the methods, while others switch to a different method due to side effects, as well as stock-outs of preferred methods in local public health care facilities (10, 31, 32). Discontinuation is an important them and should be explained. Method switching perse is not a problem, especially if a woman experiences side effects, so should be dealt with as a separate issues. Stock outs is a challenge in local health facilities and should be referenced, and explained why it’s a problem (in 	We have added the following information in the revised manuscript: ‘High rates (29%) of contraceptive users discontinue using contraception within a year of starting to use contraception, due to side effects (28%), the desire to become pregnant (19%) or the desire for a more effective method (11%) (10). Contraceptive stockouts remain high in South Africa, deterring women from accessing preferred contraceptive methods in local public health care facilities, thereby contributing to low or inconsistent uptake of contraceptives (32).’	Page 5, lines 34-39
	‘Contraceptive discontinuation has dire implications, including unplanned pregnancy. Therefore, women should be empowered to avoid unplanned	Page 19, lines

terms of SRH rights and choice). Reference 31 is out of date and more recent articles should be cited.	pregnancies, by identifying alternative contraceptive methods that meet their individual needs.'	
---	--	--

VERSION 2 – REVIEW

REVIEWER	Pleaner , Melanie University of the Witwatersrand Johannesburg Faculty of Health Sciences
REVIEW RETURNED	24-Nov-2022

GENERAL COMMENTS	comments from reviewer taken into account and included satisfactorily in the re-submission
--

REVIEWER	Larsson, Elin C. Karolinska Inst
REVIEW RETURNED	20-Dec-2022

GENERAL COMMENTS	Review- 2, revised paper 20/12 2022 General: The paper has very much improved, e.g. the English throughout, and the discussion shortened. What is now the major problem is the result section. The way the results are presented it feels like the analysis hasn't been thoroughly conducted. The aim is to look at reasons why women discontinue contraception. Pls review the results section to check if all results presented are corresponding to the aim of the paper.. Introduction Lines 86-87: "b) not postpartum amenorrhoeic, are considered fecund, want to delay their next pregnancy for two years or more...". Remove: "for 2 years or more" Lines 130-133: add where these figured are from SA? Global? Methods: Lines 154—155 talks about people on ART, but the paper has not included % HIV infected in the study setting, pls add that information. Was HIV diagnosis part of the selection of study participants, or data collection tool? Line 195- add information about the venue as where the interviews were held Results: In general
--

	reduce on the number of quotes and add more text prior to the quotes. Also check repeating points in the quotes and remove some of them. Aim to reduce the length of the results section Table 1: Add information regarding number of children 299-308: I don't understand this result. Pls provide more details in the text written before the quote or remove this section. Line 322: Starts" In addition to contraceptive mistrust", but it's the first sentence in the section so "In addition" is wrong here. Line 419: Write contraceptive method rather than contraception method. Discussion Check the language throughout
--	--

VERSION 2 – AUTHOR RESPONSE

Reviewer 2	
Comments from reviewer taken into account and included satisfactorily in the re-submission	Thank you. We appreciate your review, and we believe that it has helped to improve the quality of this manuscript.
Reviewer 3	
General: The paper has very much improved, e.g. the English throughout, and the discussion shortened. What is now the major problem is the result section. The way the results are presented it feels like the analysis hasn't been thoroughly conducted. The aim is to look at reasons why women discontinue contraception. Pls review the results section to check if all results presented are corresponding to the aim of the paper..	We note and appreciate the comment from the reviewer. We have had another look at our data, our aim and what we presented in our results section. As a result, we have made the following changes to the study title and the aim of the study. The aim has been revised to read as follows: "This study explored women's experiences of using contraceptive methods in KwaZulu-Natal, South Africa". This was the initial study aim, which is corresponding with the results presented in this study.
Introduction  • Lines 86-87: "b) not postpartum amenorrhoeic, are considered fecund, want to delay their next pregnancy for two years or more...". Remove: "for 2 years or more" 	This has been removed.

 • Lines 130-133: add where these figures are from SA? Global? 	The figures are from South Africa. This is reflected at the beginning of the sentence, which reads as follows: “In South Africa, only 46% of births occurring within a five-year period preceding the SADHS 2016, were intended at the conception time, while 34% and 20%, respectively, were mistimed and unwanted” page 6, lines 130-133
Methods:  • Lines 154—155 talks about people on ART, but the paper has not included % HIV infected in the study setting, pls add that information. 	This has been added as follows: “the eThekweni Metropolitan Municipality, which has the largest number of people on lifelong antiretroviral therapy (ART) (N=450 238) in the province, and accounts for 16.7% of HIV prevalence”. Line 145-146, page 6
 • Was HIV diagnosis part of the selection of study participants, or data collection tool? 	No, HIV diagnosis was not part of the study participants or the data collection tool.
 • Line 195- add information about the venue as where the interviews were held 	This has been added in page 6 lines 160-161: “Interviews were conducted after services had been rendered by the facility, in a designated quiet area within the facility”
Results:  • In general  o reduce on the number of quotes and add more text prior to the quotes. o Also check repeating points in the quotes and remove some of them. o Aim to reduce the length of the results section 	Noted, the number of quotes have been reduced. We have also checked repeating points and removed some of them. Overall, the length of the results section has been reduced as per the reviewer’s recommendation.
 • Table 1: Add information regarding number of children 	This information was not collected from study participants. However, we ensured that only sexually active participants formed part of this study, as indicated in lines 158-159, page 6.
 • 299-308: I don’t understand this result. Pls provide more details in the text written before the quote or remove this section. 	In line with the suggestion to reduce the number of quotes, this has been removed.

 • Line 322: Starts” In addition to contraceptive mistrust”, but it’s the first sentence in the section so “In addition” is wrong here. 	This has been deleted.
 • Line 419: Write contraceptive method rather than contraception method. 	This has been changed.
Discussion  • Check the language throughout 	The language, grammar and spelling has been checked throughout the document.